# Seropositivity for *Coxiella burnetii* in Wild Boar (*Sus scrofa*) and Red Deer (*Cervus elaphus*) in Portugal

**DOI:** 10.3390/pathogens12030421

**Published:** 2023-03-07

**Authors:** Humberto Pires, Luís Cardoso, Ana Patrícia Lopes, Maria da Conceição Fontes, Manuela Matos, Cristina Pintado, Luís Figueira, João Rodrigo Mesquita, Ana Cristina Matos, Ana Cláudia Coelho

**Affiliations:** 1Polytechnic Institute of Castelo Branco, 5200-130 Castelo Branco, Portugal; 2Animal and Veterinary Research Centre, Department of Veterinary Sciences, University of Trás-os-Montes e Alto Douro (UTAD), 5000-801 Vila Real, Portugal; 3Associate Laboratory for Animal and Veterinary Sciences (AL4AnimalS), 5000-801556 Vila Real, Portugal; 4Centre for the Research and Technology of Agro-Environmental and Biological Sciences (CITAB), UTAD, 5000-556 Vila Real, Portugal; 5Research Center for Natural Resources, Environment and Society, Polytechnic Institute of Castelo Branco, 5200-130 Castelo Branco, Portugal; 6Researcher at Q-RURAL—Quality of Life in the Rural World, Polytechnic Institute of Castelo Branco, 5200-130 Castelo Branco, Portugal; 7ICBAS—School of Medicine and Biomedical Sciences, Porto University, 4099-002 Porto, Portugal; 8Epidemiology Research Unit (EPIUnit), Instituto de Saúde Pública da Universidade do Porto, 4099-002 Porto, Portugal; 9Laboratório para a Investigação Integrativa e Translacional em Saúde Populacional (ITR), 4099-002 Porto, Portugal

**Keywords:** *Coxiella burnetii*, Portugal, Q fever, red deer, seroprevalence, wild boar

## Abstract

Q fever is caused by the pathogen *Coxiella burnetii* and is a zoonosis that naturally infects goats, sheep, and cats, but can also infect humans, birds, reptiles, or arthropods. A survey was conducted for the detection of antibodies against *C. burnetii* in a sample of 617 free-ranging wild ruminants, 358 wild boar (*Sus scrofa*) and 259 red deer (*Cervus elaphus*), in east–central Portugal during the 2016–2022 hunting seasons. Only adult animals were sampled in this study. Antibodies specific to *C. burnetii* were detected using a commercial enzyme-linked immunosorbent assay (ELISA; IDVet^®^, Montpellier, France) according to the manufacturer’s instructions. The seroprevalence of *C. burnetii* infection was 1.5% (*n* = 9; 95% confidence interval [CI]: 0.7–2.8%). Antibodies against *C. burnetii* were detected in 4/358 wild boar (1.1%; 95% CI: CI: 0.3–2.8%) and 5/259 red deer (1.9%; 0.6–4.5%). Results of the present study indicate that antibodies against *C. burnetii* were present in wild boar and red deer in Portugal. These findings can help local health authorities to focus on the problem of *C. burnetii* in wildlife and facilitate the application of a One Health approach to its prevention and control.

## 1. Introduction

*Coxiella burnetii* is a Gram-negative obligate intracellular, γ-Proteobacteria, which is the etiologic agent of Q fever, a worldwide zoonosis [1]. Based on phylogenetic investigations of the 16S rRNA, this bacterium belongs to the order Legionellales and the family Coxiellaceae. *C. burnetti* has two antigenic phases: phase I, virulent, with smooth lipopolysaccharides (LPS); phase II, avirulent, with rough LPS [2]. In wild ruminants, *C. burnetii* infection has been documented worldwide [3]. Q fever was first described in Australia in 1935 when Edward Holbrook Derrick investigated a disease in a group of abattoir workers in Brisbane, Queensland, Australia. The “Q” comes from “query” fever, as named by Derrick [4]. Q fever is present in the Iberian Peninsula [5], but little is known about its current occurrence in Portugal, its geographic distribution, or the role of wild mammals [6]. Although several studies have analyzed the epidemiology of infection, it is still poorly understood [7]. *C. burnetii* can survive in the environment for long periods under conditions of low humidity and high temperature [8], due to its ability to produce extremely resistant small, dense spores [9]. Among the main characteristics of *C. burnetii* is its resistance to physical and chemical agents. In soil, at room temperature, this bacterium may remain viable for 4 months. In tick feces, it resists for up to 36 months and is resistant to UV radiation. Concerning chemical agents, the bacterium is resistant to sodium hypochlorite solution at 100 mg/mL and to pH variations. It survives for around 3 days in 0.5% formaldehyde and 15 min in contact with 50% ethanol. The microorganism is highly resistant to heat treatment and can withstand a temperature of 60 °C for up to 30 min. Due to this characteristic, the traditional method of milk pasteurization has suffered alterations as it does not meet food safety standards. During fast pasteurization, the temperature of raw milk is maintained at 72 °C for 15 s. This method is considered to be more effective in destroying the microorganism [10]. Q fever affects various mammals, including domestic mammals, which act as a reservoir for the infection and pose a severe public health threat [11]. Small ruminants are often identified as the major contributors to the transmission of the disease to humans [12]. The main transmission route of *C. burnetii* is the aerogenic route, through inhaling aerosols or dust containing the microorganism [13]. *C. burnetii* can bind to dust particles, disperse over long distances (up to about 18 km in favorable weather conditions), and survive in adverse conditions. Environmental dispersal is a risk factor for outbreaks in humans and livestock, with wind being an important component of environmental transmission [8]. Ticks are an efficient vector of *C. burnetii*, which has been isolated from several species of ticks collected from vegetation and domestic or wild animals. Some tick genera in which *C. burnetii* has already been detected are *Amblyomma*, *Dermacentor*, *Hyalomma*, *Rhipicephalus*, and *Ixodes* [14]. The oral transmission route is less common but can occur through consumption of contaminated raw milk and its derivatives. In addition to these routes, direct contact with infected animals or contaminated fomites, vertical transmission, and sexual transmission can occur. Contact with abortion material, vaginal discharges, and mucous membranes of infected animals can also be modes of contamination [1]. Infection in humans occurs mainly through inhalation of contaminated aerosols, ingestion of raw (unpasteurized) milk and its derivatives and contact with excreta from infected animals [15]. Aerosolization of *C. burnetii* through fertilizer distribution in fields is also considered a risk factor for Q fever outbreaks in humans [8]. Transmission of *C. burnetii* is associated with abortion in domestic ruminants, and other modes of transmission may occur such as contact with infected blood or milk [16]. *C. burnetii* can be isolated from infected ruminants’ feces, milk, colostrum, urine, vaginal secretions, fetal membranes, placenta, and amniotic fluid [17]. When animals are infected, *C. burnetii* enters the body and can be located in the mammary glands, the supramammary lymph nodes, the placenta, and the uterus, and then be excreted at subsequent births [18]. Once established in the placenta, intrauterine infection may be latent or active. If it remains latent, it may be restricted solely to the placenta or spread to the fetus. Offspring are born apparently normal and may be congenitally infected or not. If active, the infection may be confined to the placenta or spread to the fetus via hematogenous or amniotic routes, in which case it causes abortion, premature delivery, stillbirths, or weak offspring. The outcome of intrauterine infection depends on several factors, such as the virulence of the bacteria, immune system of the mother and fetus, the severity of infection, damage to the placenta, gestation time, and number of infected fetuses [19]. Generally, abortions mainly result from severe lesions of the placenta, necrosis of the cotyledons, and thickening of the intercotyledonary areas [20]. *C. burnetii* causes various reproductive signs, in ruminants, such as abortions (particularly late in gestation), stillbirths, premature births, and weak neonates [12]. Dystocia can also occur due to fetal death, poor fetal positioning, or uterine inertia [19]. Placentitis, endometritis, mammary gland lesions [9], infertility, pneumonia, anorexia, depression, agalaxia, and placental retention are clinical signs that may also be present in ruminants, but are not as frequent [21]. Infected sheep and goats can suffer abortions and excrete large amounts of the bacterium into the environment in subsequent pregnancies, with goats being particularly susceptible. Infection can persist in livestock for years [9]. In humans, after an incubation period of 2 weeks, Q fever is a disease that can manifest as an acute, self-limited febrile illness with flu-like symptoms (fever, headache, myalgia, and joint pain). The period of incubation is 2 weeks. Pneumonia and hepatitis are common complications. Chronic illness rarely occurs, but is more severe and may present with endocarditis, hepatitis, fatigue syndrome, vasculitis, osteomyelitis, miscarriages, or premature births. Pregnant women are associated with high risk to chronic Q fever, which may result in miscarriage or intrauterine fetal death [22]. However, this infection is usually asymptomatic and self-limiting [21]. Clinical signs and lesions caused by *C. burnetii* infection in wildlife include reproductive failure as miscarriages, stillbirths and weak offspring, and placentitis resemble those observed in livestock [7,23,24]. Many infected animals have no clinical signs. The absence of pathognomonic clinical signs and the fact that seronegative animals can excrete bacteria make diagnosis more difficult. In addition, there are no reference diagnostic techniques, which represents a problem for detecting and surveilling cases [13]. Although there are suggestive clinical signs, none of them is pathognomonic of Q fever, and there is a list of differential diagnoses to be ruled out. In addition to this disease, differential diagnoses are campylobacteriosis, brucellosis, listeriosis, chlamydiosis (enzootic abortion), leptospirosis, and toxoplasmosis. All these diseases can cause abortions in small ruminants and are relevant to public health. Abortions can also be idiopathic due to metabolic or hormonal deficiencies, nutritional deficiencies, trauma, or poisoning [25].

The Infection and ecology of *C. burnetii* has been overlooked in wildlife, and the influence of host and environmental factors is still largely unknown, despite evidence that certain wild species behave as reservoirs of *C. burnetii* [7]. Q fever in humans is a worldwide public health problem that needs a One Health approach [26,27], since domestic ruminants are the main source of infection for humans, although wildlife can act as reservoirs. Wildlife species can shed the bacteria and contaminate the environment with *C. burnetii* and transmit it to animals and humans [3,7]. The diagnosis of Q fever is confirmed by various serological techniques for detecting of antibodies against *C. burnetii* antigens, as well as isolation of the microorganism and its genetic material. The choice of which depends on the purpose of the investigation and the types of samples investigated. The available tests can be classified into two types: direct, which aims to search for the presence of the agent (histological analysis, molecular analysis, and isolation), or indirect, which detects the antibodies produced during infection through serological analysis. *C. burnetii* DNA can be detected using real-time PCR techniques on swabs collected from vaginal mucus or in milk samples. Antibodies against *C. burnetii* can be detected using various serological tests such as immunofluorescence (IFI), enzyme-linked immunosorbent assay (ELISA), complement fixation test (CFT), and microagglutination [9]. The ELISA test is the most indicated for the detection of the disease in animals, presenting good specificity and high sensitivity. CFT is the least used due to its lower sensitivity concerning other indirect methods [28]. Serological surveys are widely applied to study the presence and distribution of infectious diseases in wild animals [29]. ELISA is the most widely chosen method for epidemiological studies in wildlife populations. Various commercial ELISA tests to detect *C. burnetii* antibodies in domestic ruminants can be used for wild ungulates, with modifications after validation [30,31]. Evidence of antibodies to *C. burnetii* was reported among various wild animal species in Spain including chamois (*Rupicapra rupicapra*), fallow deer (*Dama dama*), European wild boar (*Sus scrofa*), roe deer (*Capreolus capreolus*) [27], European mouflon (*Ovis aries musimon*), and red deer (*Cervus elaphus*) [27,32]. In red deer in Spain, antibody prevalence ranges from 1.6% to 8.4% [32]. Infection in wild boar has also been previously confirmed using molecular methods in Spain [3,33]. Determination of Q fever agent prevalence in red deer and wild boar can provide new insights about transmission dynamics of disease transmission. Information on potential pathogen exposure is necessary for monitoring the health of wildlife populations [27] in Portugal and the Iberian Peninsula.

To our best knowledge, the seroprevalence of Q fever in wild boar and red deer in central Portugal has not been reported in the literature so far. This study aimed to investigate the exposure to *C. burnetii* of wild ungulates in central Portugal, in order to provide data that could contribute to assess prevalence and distribution of Q fever at the national level.

## 2. Materials and Methods

Between 2016 and 2022, a survey for Q fever was performed on serum samples randomly obtained from free-ranging hunted wild ruminants killed by hunters, in east–central Portugal. The first hunted animals were sampled up to a total of 10 per year, at each site and in each year. Sampled municipalities included Alcafozes (*n* =16), Castelo (*n* =30), Cegonhas (*n* = 8), Crato (*n* = 46), Fratel (*n* = 32), Granja (*n* = 10), Idanha-a-Nova (*n* = 29), Lousã (*n* = 44), Marvão (*n* = 23), Mata (*n* = 41), Monforte (*n* = 10), Monte Fidalgo (*n* = 69), Niza (*n* = 26), Ponte de Sôr (*n* = 25), Portalegre (*n* = 49), Rosmaninhal (*n* = 31), Sarnadas do Ródão (*n* = 40), Tostão (*n* = 9), Vila Velha de Ródão (*n* = 64), and Vale Pouco (*n* = 15). These areas hold most of the wild ungulate population in Portugal. A total of 617 adult wild ungulates, representing two species, i.e., 358 wild boar (*S. scrofa*) and 259 red deer (*C. elaphus*), were examined. Information regarding age, sex, body condition, and location of capture, whenever available, was used to describe the distribution of seropositive individuals.

Blood samples were obtained from the heart or thoracic cavity of the animals during the hunting season. Blood was allowed to clot at environmental temperature, transported to the laboratory, and then centrifuged at 1500× *g* for 10 min, with serum samples being kept at −20 °C until testing. Sera were checked for the presence of antibodies to *C. burnetii* in multiple species, using an anti-multi-species HRP conjugate by an ELISA kit (ID Screen^®^ Q fever Indirect Multi-species; IDvet, Montpellier, France), in accordance with the manufacturer’s recommendations and following the guidelines for the interpretation of results. Sensitivity and specificity of this assay have been shown to be 100% (IDvet, according to the manufacturer’s internal validation report). Plate microwells were coated with *C. burnetii* phases I and II. Optical densities (OD) of the tested samples and positive and negative controls were measured by an ELISA plate reader at 450 nm. The OD ratio of the sample and positive control (S/P) was calculated for each sample as follows:[(OD_sample_ − OD_negative_) / (OD_positive_ − OD_negative_)] × 100(1)

Ratios were stratified as four different rising categories: samples with S/P < 40% were considered negative, samples with S/P between 40% and 50% were considered doubtful, samples with S/P between 50% and 80% were considered low positive, and samples with S/P > 80% were considered strongly positive. Any serum sample that was initially classified as “doubtful” was retested and, if resulting doubtful again, it was then considered as negative. Case definition: a wild boar or a red deer that tested positive for *C. burnetii* antibodies were considered infected.

### Statistical Analysis

The chi-square test was used to assess significant differences among the groups. A *p*-value < 0.05 was considered statistically significant. A confidence interval (CI) of 95% was calculated for all estimates by the exact binomial test.

## 3. Results

The seroprevalence of *C. burnetii* infection was 1.5% (*n* = 9; 95% confidence interval [CI]: 0.7–2.8%). Antibodies to *C. burnetii* were detected in 4/358 wild boar (1.1%; 95% CI: 0.3–2.8%) and 5/259 red deer (1.9%; 0.6–4.5%).

Of these nine positive wild ungulates, three (0.5%; 95% CI: 0.1–1.4%) were considered low positive and six (1.0%, 95% CI: 0.4–2.1%) were considered strong positive.

Regarding distribution according to municipalities, anti-*C. burnetii* antibodies were found in five of them: one wild boar from Alcafozes (6.25%; 1/16 wild ungulates), one red deer from Sarnadas de Ródão (3.6%; 1/28 wild ungulates), three red deer and two wild boar from Monte Fidalgo (7.2%; 5/69 wild ungulates), one wild boar from Rosmaninhal (3.2%; 1/31 wild ungulates), and one red deer from Vila Velha de Ródão (1.6%; 1/63 wild ungulates).

Among the positive species, prevalence in red deer (1.9%; 95% CI: 0.6–4.5%) was higher than in wild boar (1.1%; 95% CI: 0.3–2.8%), but the difference was not statistically significant (*p* = 0.303).

Serologic reactivity data according to species, sex, age, and clinical signs examined are presented in Table 1. The seroprevalence values among males and females were 0.9% (95% CI: 0.2–2.6%) and 2.1% (95% CI: 0.7–4.5%), respectively (*p* = 0.213) (Table 1). Regarding age, the lowest value of seroprevalence (0.6%; 95% CI: 0.1–1.8%) was found in juveniles, and the highest (5.1%; 95% CI: 1.1–9.1%) in adults (Table 1), but these were not statistically significant differences (*p* = 0.305). There was no significant difference in seropositivity results among clinical signs related to presence (2.5%; 95% CI: 0.5–7.3%) and absence (1.2%; 95% CI: 0.4–2.6%) in the studied species (*p* = 0.305) (Table 1).

## 4. Discussion

Q fever is caused by the pathogen *C. burnetii* and is a zoonosis whose agent naturally infects goats, sheep, and cats, but can also infect humans, birds, reptiles, or arthropods. According to the European Union (EU) annual Q fever epidemiological report for 2019, 1069 human cases were notified in the EU/European Economic Area, 958 (90%) of which were confirmed [34]. In Portugal the disease is endemic in humans with an incidence of 0.11 cases per 100,000 inhabitants, with the highest number of cases reported in the central and southern regions of the country [35].

Previous seroepidemiological studies have proven effective in investigating *C. burnetti* in wild ungulates in the Iberian Peninsula and Europe [3,23,32]. This type of investigation is fundamental for correctly designing prevention and control measures in livestock and wild ungulates under the One Health strategies.

The present study represents the largest serosurvey for *C. burnetii*, a multi-host pathogen in wild ungulates, and is the first one conducted on the prevalence of *C. burnetii* infection using commercial ELISA in Portugal, to date. The test selected to carry out this study has proven to be practical and fast compared to cultural methods that require high biosecurity conditions or to expensive molecular tests. Reports of *C. burnetii* serologically positive wild ungulates in the Iberian Peninsula include red deer (*C. elaphus*) [32,36] and wild boar (*S. scrofa*) [3]. In Portugal, only one wild mammal molecular prevalence study has been performed, but red deer and wild boar were negative to *C. burnetii* [6]. Other studies performed in Portugal have reported Q fever in domestic ruminants [6,37,38,39,40], dogs and cats [41], and humans [42,43,44,45]. Nevertheless, this study is the first report of *C. burnetii* antibodies in red deer and wild boar in east–central Portugal. Although seroprevalence seems to be low, the etiological agent appears to infect wild ungulates under study. Seropositive variation between red deer and wild boars is evidence of the existence of the recent infections and past exposures within the studied animals. Previous studies showed a high seroprevalence ranging from 8.6% to 17.9% in sheep in the country [37,46] and a seroprevalence of 37.8–61.1% in dairy cattle herds [37,38]. Interspecies contact between wild ungulates and domestic ruminants occurs in their habitat, which may favor the transmission of infectious agents such as *C. burnetii*, and the proximity of small ruminants and wild animals to humans may contribute to the transmission of the pathogen to humans [32,47]. The seropositivity found in domestic sheep in previous studies in Portugal suggests an involvement of this species on the potential *C. burnetii* spillover to other susceptible hosts such as wild ungulates.

No association was found between seroprevalence and sex, age or clinical signs. However, the high seropositivity level observed in older wild ungulates corroborated previous studies and could be explained by longer exposure to the bacterium in the environment. Studies on livestock have also found the same pattern [48,49].

In our study, seropositivity was higher in females and adults when compared with males and youngers; however, the results were not statistically significant. In a study carried out in wild boar in Montes de Toledo, south–central Spain [50], seroprevalence was higher in males (1.6%; 95% CI: 0.0–8.7%) compared to females (0.0%; 0.0–5.7%), and in adults (2.6%; 95% CI: 0.1–13.8) compared to young wild boar (0.0%; 95% CI: 0.0–26.5%).

Considering the subclinical features of Q fever, seroepidemiological studies that show the presence of infection are important in disease control, since wild ungulates can transmit the agent even while providing a seronegative result [51].

*C. burnetii* ELISA tests for livestock have been previously used to study Q fever in wild ruminants [32,47,51,52,53]. According to the manufacturer’s internal validation report, sensitivity and specificity of this assay have been shown to be 100% (IDvet^®^). However, this kit has not been validated for wildlife, and sensitivity and specificity may be lower than reported by manufacturers. The seroprevalence found in east–central Portugal, even considering individual municipalities/locations, appears to be within low ranges when compared with the prevalence observed among wild ruminants in neighboring Spain, reported to be 1.9 to 7.0% in the Basque country [3], and 6% in the Canary Islands [54]. Red deer has been identified as an important reservoir host for *C. burnetii* in the Iberian Peninsula [32,55]. In Europe, *C. burnetii* infection has previously been reported in game animals [56,57].

The dynamics of wild boar and deer population in the Iberian Peninsula is changing, with a trend toward considerable constantly growing in Portugal. This change is due to destruction and habitat fragmentation by main anthropogenic factors associated with urban expansion, agricultural practices, forestry and livestock expansion, loss of natural predators, and climate change, leading to closer proximity between livestock and domestic species and, consequently, increased interspecies contact, which facilitates the transmission of infectious diseases [58,59]. In recent decades, emerging zoonotic infectious diseases of wild animal origin have been one of the most worrying threats to human and livestock health [60]. In the Iberian Peninsula, wild ungulates are considered a reservoir of several infectious diseases such as tuberculosis, brucellosis, and paratuberculosis [61,62,63,64]. The present study contributes to the detection of potential future threats of *C. burnetii* infections from wild populations, informing on the potential involvement of wild ungulates in the infection. Wildlife monitoring is needed to identify changes in disease occurrence and measure interventions impact. This monitoring in wild ungulates allows information to be obtained to compare distribution trends and prevalence in livestock, serving as a basis for making decisions on disease control in both types of populations and as a way of assessing the effects of any disease management action [61]. The results of the present study can serve as a basis for future research by allowing the comparison of the seroprevalence in domestic and wild ungulates. It also highlights the importance of convenience sampling in providing basic descriptive information useful for the design of future epidemiological research [5]. Q fever is an example of a disease that needs the collaboration of human and animal health professionals working together in a One Health perspective to reduce the risk of infections for both humans and animals. This approach must also consider the environmental risk of Q fever associated with domestic and wild mammals, particularly in regions of nature tourism, where the human population is in close contact with countryside and, consequently, with livestock and wildlife.

## 5. Conclusions

This is the first study in wild ungulates carried out in Portugal. The results highlight the importance of a One Health, multidisciplinary approach and the integration of wild animals in the livestock the disease control of Q fever in animals and humans. Results of the present study indicated that wild boar and red deer from the center of Portugal were exposed to *C. burnetii*. There is also a public health concern, and natural reservoirs should be investigated to explain the role of wildlife in the epidemiology of infection.

## Figures and Tables

**Table 1 pathogens-12-00421-t001:** Screening for anti-*C. burnetii* antibodies in free-ranging wild animals from Portugal.

	No. Anti-*C. burnetii* Low pos./Total (%; CI *)	No. Anti-*C. burnetii* Strong pos./Total (%; CI *)	No. Anti-*C. burnetii* pos./Total (%; CI *)
**Species**			
Wild boar	3/358 (0.8%; 0.2–2.4%)	1/358 (0.3%; 0.0–1.5%)	4/ 358 (1.1%; 0.3–2.8%)
Red deer	3/259 (1.2%; 0.2–3.4%)	2/ 259 (0.8%; 0.9–2.8%)	5/ 259 (1.9%; 0.6–4.5%)
**Sex**			
Male	2/332 (0.6%; 0.0–2.2%)	1/332 (0.3%; 0.0–1.7%)	3/332 (0.9%; 0.2–2.6%)
Female	4/285 (1.4%; 0.4–3.6%)	2/285 (0.7%; 0.0–2.5%)	6/285 (2.1%, 0.7–4.5%)
**Age**			
Juvenile	2/499 (0.4%; 0.0–1.4%)	1/499 (0.2%; 0.0–1.1%)	3/499 (0.6%; 0.1–1.8%)
Adult	4/118 (3.4%; 0.9–8.5%)	2/118 (2.5%; 0.5–7.3%)	6118 (5.1%; 1.1–9.1%)
**Clinical signs**			
Absence	5/499 (1.0%; 0.3–2.3%)	1/499 (0.2%; 0.0–1.1%)	6/499 (1.2%; 0.4–2.6%)
Presence	1/118 (0.8%; 0.0–4.6%)	2/118 (1.7%; 0.2–6.0%)	3/118 (2.5%; 0.5–7.3%)

* CI, 95% confidence interval; pos., positive.

## Data Availability

The data presented in this study are available on request from the corresponding author.

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
