# Peer review of "Seropositivity for Coxiella burnetii in Wild Boar (Sus scrofa) and Red Deer (Cervus elaphus) in Portugal"

_pathogens, 2023, doi:10.3390/pathogens12030421_

Round 1

Reviewer 1 Report

The manuscrip is written very well. I have some remarks:

1. keywords: I suggest add Q fever and delate ELISA

2. The statistical analysis was performed including data about: sex, age, clinical signs but  in the section results and discussion lack is information about results of statisical analysis.

3. line 180 - I do not know what does it mean?may be is mistake?

4. In some part of manuscrit C. burnetii is written without italic (e.g. line 105).

5. Figure 1 is very general and in my opinion so is not needed.

Author Response

REVIEWER # 1

  1. The manuscript is written very well. I have some remarks:

A Many thanks for your constructive comments.

2. keywords: I suggest add Q fever and delate ELISA

A The word ELISA has been deleted and Q fever added to the manuscript.

3. The statistical analysis was performed including data about: sex, age, clinical signs but in the section results and discussion lack is information about results of statisical analysis.

A The information regarding sex, age, and clinical signs has now been added to the results and discussion section.

4. line 180 - I do not know what does it mean?may be is mistake?

A Many thanks for the comments. This is a mistake and the sentence has now been deleted.

5. In some part of manuscrit C. burnetii is written without italic (e.g. line 105).

A Many thanks for the comments. The correction has been made throughout the manuscript.

6. Figure 1 is very general and in my opinion so is not needed.

AFigure 1 has been deleted.

Reviewer 2 Report

Line 38: numerous several studies. Delete either numerous or several.

Line 85-86: I assume this test has been modified to contain protein A and G HRP conjugates and not the original secondary antibodies. I would make sure you state this in the methods because it is important.

Lines 142-143: should read seropositive variation between red dear and wild boars is clear evidence....

Author Response

REVIEWER # 2

  1. Line 38: numerous several studies. Delete either numerous or several.

AMany thanks for the comments. The correction has been made.

  1. Line 85-86: I assume this test has been modified to contain protein A and G HRP conjugates and not the original secondary antibodies. I would make sure you state this in the methods because it is important.

A The ELISA kit is an indirect multi-species ELISA for the detection of anti-Coxiella burnetii antibodies in several species, using an anti-multi-species HRP conjugates.  This information has been added to manuscript. We confirm the information about protein A and G with Idvet.

  1. Lines 142-143: should read seropositive variation between red dear and wild boars is clear evidence....

AMany thanks for the comments. The correction has been made.

Reviewer 3 Report

The authors reported a serologic survey for Coxiella burnetii infection  in free-ranging wild ruminants, wild boar and red deer from East-central Portugal during 2016-2022 hunting seasons using a commercial available kit. This is a limited study carried out in wildlife. The manuscript needs to be checked for usage of the language.

Was any sampling plan designed for collection of the samples?

Lines 70-71 and 81-82: It is mentioned that the blood was drawn from hunted animals. Later, it is mentioned that blood was drawn from the heart or thoracic cavity of the animals. In some cases, blood was drawn from thawed carcasses that had been frozen during the hunting  season.

    The authors should mention where from the blood samples were drawn??  This has to be clarified.

Italicise C. burnetii throughout the manuscript.

Line 20: : A serologic survey was conducted for antibodies to Coxiella burnetii….

Definitely, serology will measure antibodies in the samples! Rewrite the sentence.

Line 38:  numerous several studies have analysed..

Mention either numerous or several

Lines 70-71: Between 2016 and 2022, a survey for Q fever was performed on serum samples from randomly obtained from free-ranging hunted wild ruminants killed by hunters

Rewrite the sentence.

Line 82: blood was drawn from thawed carcasses that had been frozen during the hunting…

Whether blood can be drawn from dead animals or frozen animals??

Line119:  red deer prevalence…  

Prevalence in red deer…

Line 177:  Results of the present study indicate that wild boar…

Results of the present study indicated that wild boar……

Author Response

REVIEWER # 3

  1. The authors reported a serologic survey for Coxiella burnetii infection in free-ranging wild ruminants, wild boar and red deer from East-central Portugal during 2016-2022 hunting seasons using a commercial available kit. This is a limited study carried out in wildlife. The manuscript needs to be checked for usage of the language.

A Many thanks for the comments. The manuscript has been revised by a native English speaker (Mr. Kai Diprose).

  1. Was any sampling plan designed for collection of the samples?

A Information about sampling plan has been included in the manuscript.

  1. Lines 70-71 and 81-82: It is mentioned that the blood was drawn from hunted animals. Later, it is mentioned that blood was drawn from the heart or thoracic cavity of the animals. In some cases, blood was drawn from thawed carcasses that had been frozen during the hunting season. The authors should mention where from the blood samples were drawn?? This has to be clarified.

A Many thanks for the comments. The correction has been made. The blood was drawn from the heart or thoracic cavity of the hunted animals.

  1. Italicise C. burnetii throughout the manuscript.

A Many thanks for the comments. The correction has been made throughout the manuscript.

  1. Line 20: A serologic survey was conducted for antibodies to Coxiella burnetii….

Definitely, serology will measure antibodies in the samples! Rewrite the sentence.

A Many thanks for the comments. The correction has been made. A survey was conducted for the detection of antibodies against Coxiella burnetii.

  1. Line 38: numerous several studies have analysed..

Mention either numerous or several

AMany thanks for the comments. The correction has been made.

  1. Lines 70-71: Between 2016 and 2022, a survey for Q fever was performed on serum samples from randomly obtained from free-ranging hunted wild ruminants killed by hunters. Rewrite the sentence.

A Sentence has been rewritten to read as “Between 2016 and 2022, a survey for Q fever was performed on serum samples randomly obtained from free-ranging hunted wild ruminants killed by hunters”.

  1. Line 82: blood was drawn from thawed carcasses that had been frozen during the hunting… Whether blood can be drawn from dead animals or frozen animals??

A Many thanks for the comments. The correction has been made. The blood was drawn from the heart or thoracic cavity of the hunted animals (dead animals).

  1. Line119: red deer prevalence…

Prevalence in red deer…

A Many thanks for the comments. The correction has been made.

  1. Line 177: Results of the present study indicate that wild boar…

Results of the present study indicated that wild boar……

A A – Many thanks for the comments. The correction has been made.